# Combination of Medium-High-Hydrostatic-Pressure Treatment with Post-/Pre-Heat Treatment for Pasteurization of *Bacillus subtilis* Spore Suspended in Soy Milk

**DOI:** 10.3390/microorganisms13071469

**Published:** 2025-06-24

**Authors:** Morimatsu Kazuya

**Affiliations:** Department of Food Production Science, Graduate School of Agriculture, Ehime University, 3-5-7 Tarumi, Matsuyama 790-8566, Ehime, Japan; morimatsu.kazuya.nc@ehime-u.ac.jp; Tel.: +81-89-946-9967

**Keywords:** *Bacillus subtilis*, heat-resistant spore, medium/high hydrostatic pressure, germination induction

## Abstract

Medium-high-hydrostatic-pressure (MHHP) treatment can induce the spore to germinate via activating the germination receptor, subsequently resulting in the loss of the heat resistance of the spore and finally killing the germinated spore, although the ungerminated spore, even after MHHP treatment, can survive. This study aims to clarify the pasteurization effect of the combination of MHHP treatment with post-/pre-heating treatment on *Bacillus subtilis* spores suspended in soy milk as a food model. Regarding the results, the D value, as a known heat resistance indicator of the MHHP-treated spore, decreased in comparison with the untreated spore. However, the activation energies required for killing both the untreated and the MHHP-treated spores were equivalent, which indicated that the heat conductivity of the ungerminated spores might be increased by MHHP treatment. When the spore was subjected to pre-heating treatment and subsequently to MHHP treatment, the pasteurization effect of MHHP treatment differed with the pre-heating temperature. Pre-heating treatment at 80 °C could promote pasteurization, while that at 90–100 °C could suppress it, which might be caused by the heat activation/inactivation of germination receptors. From these results, the presence of post-/pre-heat treatment could be an important factor for the pasteurization of *B. subtilis* spores via MHHP treatment.

## 1. Introduction

Food loss and waste account for 38% of the total energy consumption in the global food system [1], which has been identified as a global problem for achieving the Sustainable Development Goals (SDGs). Food loss and waste occur during storage, processing, and distribution, even before it reaches consumers. One of the causes of food loss and waste is bacterial food spoilage. Generally, the shelf life of food is extended by intervention technologies. In particular, a retorting temperature above 100 °C can give food a long shelf life and allow for storage at ambient temperatures. However, retorting requires high energy costs. Thus, the SDGs call for novel technology that can kill bacteria with minimal energy.

*Bacillus* bacteria are soil-borne bacteria that can contaminate food products through air exposure or contact with raw agricultural material such as grains, meat, milk, and vegetables, particularly in food processing environments [2]. Typically, most bacteria that contaminate food products are killed during cooking at high temperatures [3]. However, since *Bacillus* bacteria form heat-resistant spores, these spores can survive cooking at high temperatures and may lead to the deterioration of food products [2,3,4,5]. To extend the shelf life of food products, controlling *Bacillus* spores is important. Generally, to reduce the risk of food deterioration caused by *Bacillus* spores, food products are subjected to retort sterilization above 100 °C [2,3,4,5]. However, extensive heat treatments cause the deterioration of food quality, including taste, color, flavor, and nutritional value, and have high energy costs [6]. Thus, novel technology that can kill *Bacillus* spores at lower temperatures during heat treatments is required to minimize energy costs and the heat deterioration of food quality in order to achieve sustainable development.

Although *Bacillus* spores exhibit high resistance to various stressors, such as not only heat but also pressure, UV-C, oxidants, and more, the germinated spores lose all of their high-resistance characteristics [2,4,5,7,8]. In this study, medium-/high-hydrostatic-pressure (MHHP) treatment at 100 MPa is used as a nonthermal inducible method of *Bacillus* spore germination. Many researchers have investigated the MHHP germination induction of *B. subtilis* spores and partially clarified a potential mechanism [7,8,9,10,11,12,13]. In the MHHP germination induction of *B. subtilis* spores, similarly to a pathway of nutritional germination, germination receptors are activated, and the spores begin to germinate when they are subjected to MHHP treatment [7,8,9,10,11,12,13]. The germinated spores undergoing MHHP treatment lose their high-resistance characteristics, which can be killed immediately, even by heat treatments at 60–70 °C [8,10,11,12]. Thus, the MHHP germination technology can minimize the heat deterioration of the food quality and the energy cost. However, in past studies, it was clarified that the MHHP germination technology cannot germinate all of the spores, and an ungerminated spore after MHHP treatment is called a super-dormant spore [8,10,11,13]. Therefore, to kill all of the spores, MHHP treatment and post-heat treatments at high temperatures that can kill the spores must be carried out. Additionally, in general food processing, cooking at high temperatures is conducted before the pasteurization process. However, the effect of pre-heating on MHHP treatment in the pasteurization process has been unclear.

Against this background, this study aims to investigate the pasteurization of *Bacillus subtilis* spores suspended in soy milk as a food model via a combination of MHHP treatments with pre-/post-heat treatments and to clarify the effect of post-/pre-heating treatments on the pasteurization of *B. subtilis* spores via MHHP germination induction.

## 2. Materials and Methods

### 2.1. Bacterial Strain and Growth Conditions

*Bacillus subtilis* NBRC 111470, known as the 168 strain, obtained from the National Institute of Technology and Evaluation, was used in this study. The strain was maintained at −80 °C in a 0.9% saline solution containing 20% glycerol. After thawing, the bacterial solution was inoculated in 20 mL of a sporulation medium (pH 7.4) that contained 0.8% Nutrient Broth (Difco TM Nutrient Broth, Becton, Dickinson and Company, Franklin Lakes, NJ, USA), 2.0 mM MgCl_2_, 1 mM CaCl_2_, 0.01 mM MnCl_2_, 1 mM FeCl_3_, and 27.0 mM KCl (Wako Pure Chemical Industries, Ltd., Osaka, Japan). The culture was incubated at 37 °C with 130 rpm of agitation for 24 h.

### 2.2. Plant Material Sample Preparation

After incubation, the culture was heated at 65 °C for 30 min in order to kill the vegetative cells. Spores were collected by centrifugation (3500 rpm, 10 min, 25 °C), and the resulting pellet was then resuspended in 40 mL of sterile distilled water. This procedure was repeated twice. After three rounds of centrifugation, the resulting pellet was resuspended in 5 mL of sterile distilled water. Then, the resuspension was heated at 65 °C for 30 min to kill the vegetative cells. Subsequently, the spore resuspension was inoculated with a 9-fold amount of soy milk, which was purchased in a retail location. The spore solution of 1.5 mL was heat-sealed into each of the sterile polystyrene bags (80 mm × 80 mm) in order to utilize it for MHHP treatments and pre-/post-heat treatments.

### 2.3. Pre-/Post-Heat Treatment

Each of the sample bags was subjected to immersion in a water bath (SB-1300, Tokyo Rikakikai Co., Ltd., Tokyo, Japan) at 80–90 °C and boiling at 100 °C for the pre-/post-heat treatment. After pre-/post-heat treatments, the heated sample was immediately chilled with tap water. Pre-/post-heat-treatment temperature was measured by a K-type thermocouple thermometer (S270WP-01, Tokyo, Sato Keiryoki Mfg. Co., Ltd., Japan).

### 2.4. Medium-/High-Hydrostatic-Pressure Treatment

MHHP treatment was carried out by a high-pressure food processor (Marugoto Ekisu EFS-0.5L, Toyo Kouatsu, Ltd., Hiroshima, Japan) using water as a pressure medium. The pressure medium temperature was maintained at 65 °C by an internal heater inside the pressure chamber (Φ 80 mm × D 140 mm). The untreated sample and pre-heated sample were immersed in the pressure chamber and pressurized up to 100 MPa at 1.1 MPa/s, and this pressurization was maintained for 30 min. Then, the bag at 100 MPa was depressurized to ambient pressure at 10 MPa/s. After decompression, the MHHP-treated sample was immediately chilled with tap water.

### 2.5. Bacterial Count and Calculation of D Value, Z Value, Activation Energy, and Frequency Factor

The population of *B. subtilis* spores in all of the samples was determined by a direct plating method. Samples were serially diluted in sterile saline water, then a diluted solution of 0.1 mL was plated in duplicate on standard methods agar (Nissui Pharmaceutical Co., Ltd., Tokyo, Japan), and the plates were incubated at 37 °C for 24 h. The population of *B. subtilis* spores was calculated from the number of colonies formed on the plate. The minimum detection level of the *B. subtilis* spore population was 1.0 log CFU/mL using the direct plating method, and no detection (N.D.) was less than 1.0 log CFU/mL.

The D value is the holding time required to reduce the bacterial count to one-tenth during the maintenance of the target post-heat temperature, and the Z value is the elevation temperature required to reduce the D value to one-tenth; both are known as indicators of bacterial heat resistance. The D value is the reciprocal of the slope of a decreasing common logarithmic straight line in a graph showing the relationship between post-heating time and bacterial number. To calculate the D value at each of the post-heat-treatment temperatures, the bacterial number of the four samples for different post-heat-treatment times was approximated to a decreasing common logarithmic straight line utilizing the least squares method. The Z value is the reciprocal of the slope of a decreasing common logarithmic straight line in a graph showing the relationship between post-heating temperature and the D value. For calculating the Z value, three data points of the D value at each of the post-heat-treatment temperatures were approximated to a decreasing common logarithmic straight line. Additionally, the activation energy (*E_a_* (J·mol^−1^)) and frequency factor (*a*) for the heat-killing reaction of bacteria are calculated. As shown in Equation (1), the Arrhenius equation, which explains the relationship between chemical reaction rates, such as the heat-killing reaction rate of bacteria with different reaction temperatures, contains the activation energy and frequency factor as indefinite variables. The activation energy and frequency factor are defined as the energy required for the chemical reaction and the frequency with which the chemical reaction occurs. The rate constant of the bacterial inactivation (*k*) at each of the post-heat-treatment temperatures is the slope of a decreasing natural logarithmic straight line in a graph showing the relationship between post-heating time and bacterial number. For calculating *E_a_* and *a*, three data points of *k* at each of the post-heat-treatment temperatures were approximated to a decreasing natural logarithmic straight line in a graph showing the relationship between the absolute temperature of the post-heat treatment and the rate constant of bacterial inactivation (*k*).(1)$${log}_{e}k=-\frac{1}{RT}\times E_{a}+{log}_{e}a$$
Here, *R* is the gas constant of 8.3 J·mol^−1^·K^−1^, and *T* is the absolute temperature of the heat treatment (K).

### 2.6. Statistical Analysis

All experiments were conducted three times independently. Experimental results were expressed as mean ± standard deviation. When comparing the experimental results, statistical significance was verified by the Tukey–Kramer method (*p* < 0.05).

## 3. Results and Discussion

### 3.1. Pasteurization of B. subtilis Spore by Combination of Post-Heat Treatment With and Without MHHP Treatment

Figure 1 shows the pasteurization of *B. subtilis* spores in combination with post-heat treatments at 80 °C, 87 °C, and 100 °C with and without MHHP treatment. MHHP treatment for 30 min decreased the viable bacterial count of *B. subtilis* spores from 7.4 ± 0.1 to 3.6 ± 0.2 log CFU/mL, which was the initial bacterial count of untreated spores and MHHP-treated spores before post-heat treatments, as shown by ● and ■ in Figure 1, respectively. Post-heat treatments at 80 °C for 30 min to 180 min caused a decrease from 7.1 ± 0.1 to 6.6 ± 0.1 log CFU/mL in the untreated spore and from 3.7 ± 0.1 to 2.6 ± 0.3 log CFU/mL in the MHHP-treated spore, as shown in Figure 1a. Post-heat treatments at 87 °C for 5 min to 30 min caused a decrease from 7.1 ± 0.1 to 5.9 ± 0.1 log CFU/mL in the untreated spore and from 3.4 ± 0.2 to 1.1 ± 0.5 log CFU/mL in the MHHP-treated spore, as shown in Figure 1b, respectively. Post-heat treatments at 100 °C for 0.5 min to 2 min caused a decrease from 6.9 ± 0.0 to 4.4 ± 0.1 log CFU/mL in the untreated spore and from 3.0 ± 0.1 log CFU/mL to no detection (<1.0 log CFU/mL) in the MHHP-treated spore, as shown in Figure 1c. A decrease in the bacterial count was linear with the extension of post-heating time, regardless of post-heating temperature. Each of these linear lines was approximated utilizing the least squares method, as shown by the dotted line and the solid line for the untreated spore and the MHHP-treated spore in Figure 1. From the slopes of these variables, the D value (min), Z value (°C), activation energy (*E_a_* (J·mol^−1^)), and frequency factor (*a*) were calculated.

Table 1 shows the D value, Z value, activation energy, and frequency factor in *B. subtilis* subjected to post-heat treatments of its untreated spores and MHHP-treated spores, calculated from experimental results of three independent trials. The D value of the MHHP-treated spore was significantly less than that of the untreated spore. Ogino and Nishiumi reported that MHHP treatment at 200 MPa introduced moisture into *B. subtilis* spores [14]. Thus, MHHP at 100 MPa could also introduce moisture into the spore, which might result in an increase in heat conductivity of the spore and a decrease in the D value. On the other hand, the Z value, the activation energy, and the frequency factor showed no significant difference between the untreated and MHHP-treated spores. Therefore, the activation energy required for heat-killing the spore and the frequency factor for the heat-killing reaction of the spore could be constant regardless of whether it was for the untreated or MHHP-treated spore. From these results, when *B. subtilis* spores were subjected to MHHP treatment and post-heat treatment, MHHP treatment could only decrease the amount of spores resistant to post-heat treatment at high temperatures.

### 3.2. Germination Induction Effect of Combination of MHHP Treatment With and Without Pre-Heat Treatment of B. subtilis Spore

The effect of pre-heat treatments on the germination induction of the *B. subtilis* spore by MHHP treatment was investigated by comparing the pasteurization of the spore between MHHP treatment alone and the combination treatment of MHHP treatment immediately after pre-heat treatment. Figure 2 shows the pasteurization effect of MHHP treatment on *B. subtilis*’s untreated spores and pre-heated spores at 80 °C, 90 °C, and 100 °C for 1–4 min. As shown by “■” in Figure 2, pre-heating at 80–90 °C for 4 min decreased the viable count of the spore to less than a one-logarithm reduction, although pre-heating at 100 °C decreased the viable count drastically with extended pre-heating times. Pasteurization of the spore by MHHP treatment alone without pre-heating caused a logarithmic decrease of 3.6–3.9, which was calculated by comparing the viable bacterial count between the “■” value of the untreated spore and the “□” value of the MHHP-treated spore. The pasteurization effect of MHHP treatment on the pre-heated spore at all temperatures was calculated by comparing the viable bacterial count between the “■” value of the pre-heated spore and the “□” value of the MHHP-treated spore. The pasteurization effect of MHHP treatment on the pre-heated spore at 80 °C for 1–4 min demonstrated logarithmic reductions of 3.6, 3.5, 3.5, and 3.4, respectively. The pasteurization effect of MHHP treatment on the pre-heated spore at 90 °C for 1–4 min indicated logarithmic reductions of 3.3, 3.1, 3.0, and 2.9, respectively. The pasteurization effect of MHHP treatment on the pre-heated spore at 100 °C for 1–4 min indicated logarithmic reductions of 2.2, 1.1, 0.5, and 0.1, respectively. Therefore, pre-heating and extended pre-heating times suppressed the pasteurization effect of MHHP treatment. In particular, pre-heating at 100 °C suppressed the pasteurization effect considerably. However, since pre-heating at 80 °C and 90 °C did not kill most of the spores, the effect of pre-heating at 80 °C and 90 °C on the pasteurization of the spores by MHHP treatment is insignificant.

Thus, when the killing ratio of the spore by pre-heating was equivalent, the pasteurization of the *B. subtilis* spore by MHHP treatment was compared among the pre-heated spores at 80 °C for 420 min, 90 °C for 30 min, and 100 °C for 1.5 min, as shown in Figure 3. Pre-heating decreased from 7.4 ± 0.1 log CFU/mL to 5.6 ± 0.1 log CFU/mL at 80 °C for 420 min, to 5.8 ± 0.1 log CFU/mL at 90 °C for 30 min, and to 5.7 ± 0.5 log CFU/mL at 100 °C for 1.5 min. The pasteurization effect values of MHHP treatment on the pre-heated spore at 80 °C, 90 °C, and 100 °C were logarithmic reductions of 4.4, 2.8, and 1.4. Therefore, compared with the pasteurization effect of MHHP treatment on the untreated spore showing a logarithmic reduction of 3.7, the pre-heat treatment at 80 °C increased the pasteurization effect, while pre-heating at 90 °C and 100 °C decreased it. The pasteurization by MHHP treatment can be achieved via the MHHP germination induction of the spore [7,8,9,10,11,12,13]. During the process of the MHHP germination induction of the spore, MHHP treatment can firstly stimulate germination receptors, and the germination can progress [7,8,9,10,11,12,13]. Additionally, heating at 60–80 °C can activate germination receptors and promote the germination of the spore, while heating at higher temperatures can inactivate germination receptors and suppress germination of the spore [14,15,16,17]. Therefore, a difference in the effect of pre-heating temperature on MHHP pasteurization could be caused by the activation/inactivation of germination receptors, where the threshold of pre-heating temperature needed to activate/inactivate MHHP pasteurization could be around 80 °C.

### 3.3. Effect of Pre-Heat Treatment on Decrease in Heat Resistance of B. subtilis Spore by MHHP Treatment

Table 2 shows the effect of the pre-heat treatment at 100 °C and MHHP treatment on the heat resistance of *B. subtilis* spores with post-heat treatments at 87 °C. Although the pre-heat treatment at 100 °C suppressed the pasteurization of *B. subtilis* spores by MHHP treatment via MHHP germination, it was unclear whether the pre-heat treatment could suppress the decrease in the heat resistance of *B. subtilis* spores by MHHP treatment, which was a focus of subsequent investigations. The D values of the untreated spore, the pre-heated spore, the MHHP-treated spore, and the combination-treated spore with pre-heating and MHHP treatment were 23 ± 4 min, 29 ± 4 min, 11 ± 2 min, and 14 ± 1 min, respectively. Significant differences were observed between the spores with and without MHHP treatment, regardless of pre-heat treatment (*p* < 0.05). Thus, pre-heat treatments could have no significant effect on the decrease in the heat resistance of *B. subtilis* spores by MHHP treatment. However, the pre-heated spores tended to increase the D value regardless of MHHP treatment. Although heat shock at 100 °C might increase the heat resistance of the spore at 87 °C, the mechanism was unclear. Further studies focusing on the effect of heat shock on the heat resistance of *B. subtilis* spores are required.

## 4. Conclusions

In this study, a combination of MHHP treatment with post-/pre-heat treatment for the pasteurization of *B. subtilis* spores suspended in a food model of soy milk was investigated. Firstly, when the MHHP-treated spore was subjected to post-heat treatment, the D value, known as an indicator of heat resistance of the MHHP-treated spore, decreased in comparison with the untreated spore, although the Z value, activation energy, and frequency factor of the untreated and MHHP-treated spores were equivalent. This result indicated that a combination of MHHP treatment with post-heat treatment could shorten the heat treatment time required for the sterilization of *B. subtilis* spores. Secondly, when the pre-heated spore was subjected to MHHP treatment, in comparison with the untreated spore, the pasteurization effect of MHHP treatment via MHHP germination induction was affected by pre-heating temperature. Pre-heat treatment at 80 °C could promote the pasteurization, while that at 90–100 °C could suppress the pasteurization, which might be caused by the heat activation/inactivation of germination receptors. However, pre-heat treatment at 100 °C had no effect on the decrease in resistance of the spore to post-heat treatment by MHHP treatment. Thus, pre-heat treatments at 100 °C caused the spore to suppress the complete loss of heat resistance and not affect the decrease in resistance. Further study is required to investigate the effect of pre-heat treatment at 100 °C on the spore. In conclusion, applying a combination of MHHP treatment with pre-/post-heat treatment in food processing can minimize the energy cost of pasteurization and decrease food loss caused by bacterial decay, which can help achieve the SDGs.

## Figures and Tables

**Figure 1 microorganisms-13-01469-f001:**
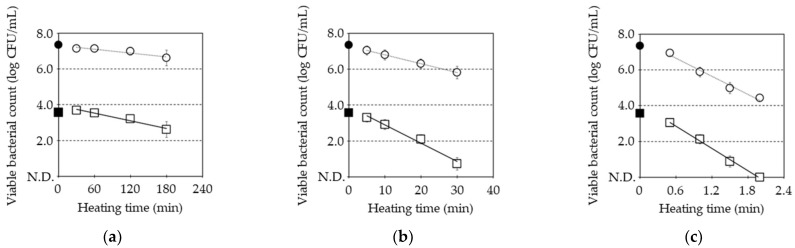
Decreases in the viable bacterial count of *B. subtilis* spores via post-heat treatments at 80 °C (**a**), 87 °C (**b**), and 100 °C (**c**) without and with MHHP treatment. The initial bacterial count of the untreated spore is indicated by “●”, and the MHHP-treated spore is indicated by “■” before post-heat treatment. The viable bacterial count of the untreated spore is denoted by “○”, and the MHHP-treated spore is denoted by “□” during post-heat treatment. The decreasing linear line of the untreated spore is indicated by the “dotted line”, and the MHHP-treated spore is indicated by the “solid line” during post-heat treatment. The plot and error bar show the mean and standard deviation calculated from the experimental data of three trials. N.D. means no detection level of viable bacterial count which was less than 1.0 log CFU/mL.

**Figure 2 microorganisms-13-01469-f002:**
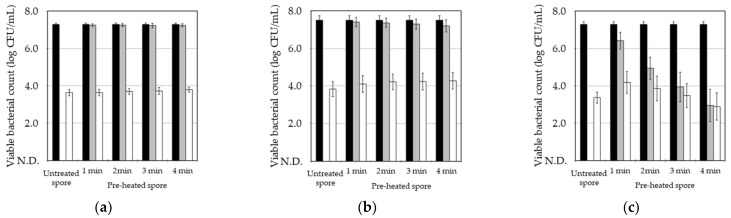
The pasteurization effect via MHHP treatment on *B. subtilis* of the untreated spore and pre-heated spore at 80 °C (**a**), 90 °C (**b**), and 100 °C (**c**). The viable bacterial count of the untreated spore is indicated by “■”, the pre-heat-treated spore is indicated by “■”, and the MHHP-treated spore is indicated by “□”. The plot and error bar show the mean and standard deviation calculated from experimental data of three trials. N.D. means no detection level of viable bacterial count which was less than 1.0 log CFU/mL.

**Figure 3 microorganisms-13-01469-f003:**
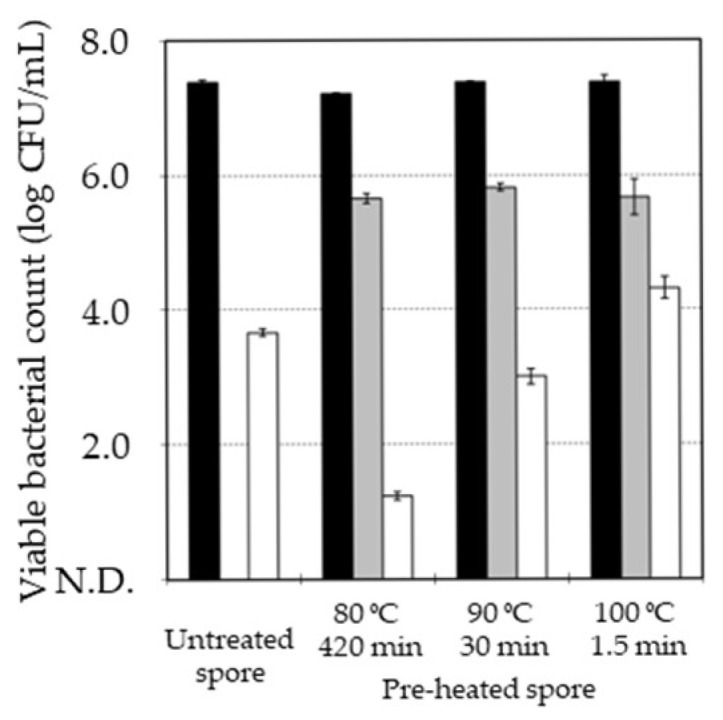
The viable bacterial count of the *B. subtilis* spore before and after MHHP treatment with and without pre-heat treatment at 80 °C for 420 min, 90 °C for 30 min, and 100 °C for 1.5 min. The viable bacterial count of the untreated spore is indicated by “■”, the pre-heat-treated spore is indicated by “■”, and the MHHP-treated spore is indicated by “□”. The plot and error bar show the mean and standard deviation calculated from experimental data of three trials. N.D. means no detection level of viable bacterial count which was less than 1.0 log CFU/mL.

**Table 1 microorganisms-13-01469-t001:** The D value, Z value, activation energy, and frequency factor of *B. subtilis* subjected to post-heat treatments of its untreated spores and MHHP-treated spores. These results show mean ± standard deviation calculated from experimental data of three trials. Significant differences between the untreated and MHHP-treated spores were evaluated by the Tukey–Kramer method (*p* < 0.05). The different letters indicate a significant difference.

	D Value (Min)			Z Value (°C)	Activation Energy (J·mol^−1^)	Frequency Factor $\left({log}_{e}a\right)$
	80 °C	87 °C	100 °C			
Untreated	300 ± 76 ^a^	20 ± 3 ^a^	0.6 ± 0.0 ^a^	7.5 ± 0.3 ^a^	335 ± 14 ^a^	109 ± 4.7 ^a^
MHHP-treated	150 ± 41 ^b^	10 ± 2 ^b^	0.5 ± 0.0 ^b^	8.2 ± 0.5 ^a^	309 ± 17 ^a^	101 ± 5.7 ^a^

**Table 2 microorganisms-13-01469-t002:** Effect of pre-heat treatment at 100 °C and MHHP treatment on heat resistance of *B. subtilis* spore with post-heat treatment at 87 °C. These results show mean ± standard deviation calculated from experimental data of three trials. Significant differences among all of the spores were evaluated by the Tukey–Kramer method (*p* < 0.05). The different letters indicate a significant difference.

D Value (min)		Pre-Heat Treatment	
		Untreated	Pre-Heated
MHHP treatment	Untreated	23 ± 4 ^a^	29 ± 4 ^a^
	MHHP-treated	11 ± 2 ^b^	14 ± 1 ^b^

## Data Availability

The original contributions presented in this study are included in the article. Further inquiries can be directed to the corresponding author.

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
