# Peer review of "Combination of Medium-High-Hydrostatic-Pressure Treatment with Post-/Pre-Heat Treatment for Pasteurization of Bacillus subtilis Spore Suspended in Soy Milk"

_microorganisms, 2025, doi:10.3390/microorganisms13071469_

Round 1
Reviewer 1 Report (Previous Reviewer 1)
Comments and Suggestions for Authors
The authors have addressed my comments regarding their original submission (#3373641)
Author Response
The authors have addressed my comments regarding their original submission (#3373641)
Thank your indication for revising the manuscript.
Reviewer 2 Report (Previous Reviewer 2)
Comments and Suggestions for Authors
Although the topic is important for food safety and the author has improved the text with major changes there are still inconsistencies and mistakes and therefore I am afraid it should be rejected for publication.
Line 196: “D value of the untreated spore was significant less than that of the MHHP-treated spore, respectively”. This phrase which describes one of the most important findings of the study is wrong because it is actually “significant higher” as correctly shown in Table 1. Although this mistake may originate from a “bad translation” in the English language, it is not excused in a revised text. The single author of this study should have advised some other experts before submitting the final version because I do not believe it is the role of the reviewer to correct elementary mistakes as these.
Figure 1: It would be more convenient for the reader to see a small legend in each diagram e.g. the temperature. The same for Figure 2
Lines 239-240: in figure 2 it is not clear what the “preheat-treated spore” is because it is not mentioned in the text and maybe it is confusing because there is a black column in every minute
Lines 311-313: “While, since pre-heat treatment at 100 °C did not prevent a decrease in resistance of the spore to post-heat treatment by MHHP treatment, a decrease in resistance of the spore to post-heat treatment by MHHP treatment might be induced not via MHHP germination induction”, this sentence should be more clear and may be supported by the values obtained in the various treatments.
General comments: more references should be used to explain the mechanisms of spore germination and how pasteurization affects this phenomenon
Comments on the Quality of English LanguageEditing is needed throughout the text
Author Response
Although the topic is important for food safety and the author has improved the text with major changes there are still inconsistencies and mistakes and therefore I am afraid it should be rejected for publication.
Line 196: “D value of the untreated spore was significant less than that of the MHHP-treated spore, respectively”. This phrase which describes one of the most important findings of the study is wrong because it is actually “significant higher” as correctly shown in Table 1. Although this mistake may originate from a “bad translation” in the English language, it is not excused in a revised text. The single author of this study should have advised some other experts before submitting the final version because I do not believe it is the role of the reviewer to correct elementary mistakes as these.
I’m sorry for my careless check, and thank your indication.
I will be more careful.
Figure 1: It would be more convenient for the reader to see a small legend in each diagram e.g. the temperature. The same for Figure 2
Thank your suggestion
Figures were revised.
Lines 239-240: in figure 2 it is not clear what the “preheat-treated spore” is because it is not mentioned in the text and maybe it is confusing because there is a black column in every minute
Thank your indication.
These sentences revised to focus difference between “preheat-treated spore” and the others.
Lines 311-313: “While, since pre-heat treatment at 100 °C did not prevent a decrease in resistance of the spore to post-heat treatment by MHHP treatment, a decrease in resistance of the spore to post-heat treatment by MHHP treatment might be induced not via MHHP germination induction”, this sentence should be more clear and may be supported by the values obtained in the various treatments.
Thank your indication.
These sentences were revised clearly.
General comments: more references should be used to explain the mechanisms of spore germination and how pasteurization affects this phenomenon
I’m sorry to submit the manuscript with few references.
However, actually, the researches focus on high hydrostatic pressure germination induction of the spore were few.
Thus, the manuscript has as many references as possible.
Round 2
Reviewer 2 Report (Previous Reviewer 2)
Comments and Suggestions for Authors
Major revisions have been made
This manuscript is a resubmission of an earlier submission. The following is a list of the peer review reports and author responses from that submission.
Round 1
Reviewer 1 Report
Comments and Suggestions for Authors
This paper is hard to read, strongly suggest having it reviewed for the English language.
It is not clear how many independent microbiological experiments were performed. If only 1 experiment, then these results are highly speculative and this ought to be stated in the abstract and conclusions.
The results are hard to understand for many reasons, the first being that the main responses of interest, D and Z values, are never defined. Secondly, the D-values appear to be reported incorrectedly for the untreated control in Table 1.
Statistical methods are not described well, missing crucial details, e.g. it is not clear how the data inform equation 1. The error bars in Tables and Figures are never defined.
Technical comments:
1. Give more details about the untreated (unheated controls). I think these are visualized in Fig 1 and 2 at time = 0
2. Line 99, define D-value, not all readers are familiar with this. I assume it is the time and temperature combination that achieves a 1-log reduction
3. Lines 99 – 101, show the linear regression equation that clearly shows how the CFU data depend on temperature and/or time. I think that k is estimated from this regression, make this clear.
4. Line 101, define Z value
5. Line 106 be clear how the data inform the equation (1) that I think is used to calculate Ea. I think that k is from the regression described in lines 99-101. Where does a come from?
6. Figure 1 shows non-detects. What is the limit of detection for the assay? Figure 1 suggests it is about log(CFU/mL) = 1.
7. lines 122-124, the decrease was not linear, Fig 1 clearly shows a non-linear relationship. However, it may be fine that you approximate this non-linear relationship with a line.
8. Table 1, because you haven’t defined D-value, it is hard to understand what you are reporting here. However, if I use the common definition of the time at which there is a 1 log reduction in CFU, then Fig 1 suggests that D=180min for the untreated at 80C, and D=10min for the untreated at 87C, please explain.
9. Table 1, explain what the value after the +/- mean.
10. Captions to Fig 2 and 3 never explain what the gray bars are.
11. Strongly recommend reporting unusual representation of negative ‘log decrease’ in Fig 2 as a positive log reduction
12. What do the error bars represent in Fig 1 and 2.
13. Line 194, second Table 1
Comments on the Quality of English LanguagePoor
Reviewer 2 Report
Comments and Suggestions for Authors
This study evaluates the effect of heat applied post- and pre- MHHP treatment on pasteurization of B. subtilis spores. Although the subject is interesting and it is always useful to provide data on alternative novel methods for eliminating bacterial spores from foods, the study is more like a preliminary short technical report written in a sloppy way than a full scientific paper. Furthermore, the methodology, the results and the discussion should be better analyzed and with more clarifications about the possible mechanisms involved. Also, the text should be revised thoroughly for grammatical and syntactical errors. In some cases, it is difficult to follow the meaning of the sentences. For these reasons, the manuscript could not be accepted for publication.
For example in the abstract some errors include the following:
Line 11: “can be survived”
Line 19: “was differed to the”
Lines 19-21: something is missing in the following sentence: “Pre-heat treatment at 80 °C promote the pasteurization while that at 90-100 °C, which might be caused by heat activation/inactivation of germination receptors”
Line 22: “was limited lower temperature”
Some other points:
Paragraph 2.3. Pre-/post-heat treatment: this paragraph should be explained better, eg. What were the exact temperatures tested, how many samples were tested overall. Moreover, by the “pre/post-heat treatment” the author probably means pre-MHHP heat treatment and post-MHHP heat treatment but the way these terms are used in the text is quite confusing.
Lines 113-114: “Initial bacterial count of B. subtilis spore treated with and without MHHP treatment for 30 min was 7.4±0.1 and 3.6±0.2 log CFU/mL.” is it the other way round?
Lines 152-153: “However, since pre-heating at 100 °C inactivated the spore considerably, inactivation of spore might have an effect on the pasteurization by MHHP treatment”, This is confusing since inactivation of the spore would mean a great log reduction but only a slight reduction was observed at 100oC
Figures 2 and 3: in the legend there should be a gray square to indicate 90oC
Line 160: “Therefore, since pasteurization effect was 3.7 of logarithm reduction for the untreated”, in line 144 the reduction was mentioned as 3.9.
Conclusion: the whole paragraph should be re-written for more clarity.
Comments on the Quality of English LanguageThere are serious grammatical and syntactical errors, the text should be thoroughly revised.
Reviewer 3 Report
Comments and Suggestions for Authors
Authors,
You have articulated your objectives for the Effect of Pre-/Post-heat Treatment on Pasteurization of Bacillus subtilis Spore Suspended in Soy Milk by Medium High Hydro-static Pressure Treatment perfectly. The clarity and relevance of your aims set a strong foundation for the study. However, I would like to suggest the following corrections to enhance your paper:
Abstract: Ok.
Introduction
I did not like your initial introduction; it is hard to understand your English.
Line 28 – 30 – Write like that, Bacillus bacteria are soil-borne microorganisms that can contaminate food products through air exposure or contact with raw agricultural materials, such as grains, meat, milk, and vegetables, particularly in food processing environments.
Line 30-31 Write like that, Typically, most bacteria that contaminate food products are killed during cooking at high temperatures.
Line 31-32 - Write like that, However, since Bacillus bacteria form heat-resistant spores, these spores can survive cooking at high temperatures and may lead to the deterioration of food products.
I would recommend incorporating references to the Sustainable Development Goals (SDGs) to strengthen the broader significance of your work. For example, highlighting SDG 3 (Good Health and Well-Being) could underscore the importance of mitigating foodborne illnesses associated with spores, particularly in vulnerable populations. This would emphasize the relevance of your research in the context of public health. Similarly, referencing SDG 12 (Responsible Consumption and Production) could align your work on sustainable sanitization practices with global priorities for eco-friendly and responsible food production. These connections would provide a wider context for your study, reinforcing its relevance beyond the immediate scope of food safety. By framing your research within the broader goals of global sustainability and health, your introduction will be stronger and offer greater clarity on the significance of your work.
2. Materials and Methods
- All ml in the paper replace for mL.
Results and Discussion
3.1 Line 111- What is conclusions? Don’t insert this word.
Conclusion: ok